# Electrochemical Evaluation of Stress Corrosion Cracking Susceptibility of Ti-6Al-3Nb-2Zr-1Mo Alloy Welded Joint in Simulated Deep-Sea Environment

**DOI:** 10.3390/ma15093201

**Published:** 2022-04-28

**Authors:** Haochen Liu, Xuehan Bai, Zhen Li, Lin Fan, Junlei Tang, Bing Lin, Yingying Wang, Mingxian Sun

**Affiliations:** 1State Key Laboratory for Marine Corrosion and Protection, Luoyang Ship Material Research Institute (LSMRI), Qingdao 266237, China; lhclhc0508@126.com (H.L.); baixuehan@sunrui.net (X.B.); lizhen@sunrui.net (Z.L.); sunmx@sunrui.net (M.S.); 2College of Chemical Engineering, Southwest Petroleum University, Chengdu 610500, China; tangjunlei@126.com (J.T.); h1900@foxmail.com (B.L.); 3Key Laboratory of Optoelectronic Chemical Materials and Devices (Ministry of Education), Jianghan University, Wuhan 430056, China

**Keywords:** titanium alloy, welded joint, stress corrosion cracking, electrochemical behavior, deep sea

## Abstract

Titanium alloys have high specific strength and excellent corrosion resistance and have been applied in deep-sea engineering fields. However, stress corrosion cracking may become one of the biggest threats to the service safety of a high-strength titanium alloy, as well as its weldment. In this work, stress corrosion cracking of a gas-tungsten-arc-welded Ti-6Al-3Nb-2Zr-1Mo (Ti6321) alloy influenced by the applied potentials in simulated deep-sea and shallow-sea environments was investigated by combining slow strain rate testing with electrochemical measurements. The results showed that the service environment and applied potential have a substantial effect on the stress corrosion cracking behavior of the Ti6321 welded joint. The Ti6321 welded joint exhibited higher stress corrosion susceptibility in a simulated deep-sea environment and at a strong polarization level owing to the diminishing protection of the passive film under passivation inhibition and the enhancement of the hydrogen effect. The fracture of a Ti6321 welded joint in the weld material could be attributed to the softening effect of the thick secondary α within the coarse-grained martensite. The electrochemical evaluation model of stress corrosion cracking susceptibility of a Ti6321 welded joint in a simulated marine environment was established by adding the criterion in the passivation region based on the literature model, and four potential regions corresponding to different stress corrosion cracking mechanisms were classified and discussed. Our study provides useful guidance for the deep-sea engineering applications of Ti6321 alloys and a rapid assessment method of stress corrosion risk.

## 1. Introduction

Titanium (Ti) alloys are a potential structural steel substitute material with high specific strength and general corrosion resistance and are being applied in an increasing number of engineering fields. In particular, Ti alloys are considered an alternative material for deep-sea use. It has been acknowledged that the general corrosion of conventional materials slows down, whereas the tendency of localized corrosion increases. Stress corrosion cracking (SCC) is the decay of the mechanical characteristics of sensitive materials under the mixed effects of stress and corrosive media [1]. As a typical localized failure mode, SCC usually occurs unexpectedly, leading to catastrophic consequences. Therefore, SCC could still be one of the main threats to the service safety of high-strength Ti alloys, as well as their weldment. Barella et al. [2] found that Ti-6Al-4V suffered from a detrimental effect related to SCC within a synthetic seawater environment, the chloride concentration and applied deformation rate having a considerable effect on the SCC toughness. Pazhanivel et al. [3] indicated that the dissolution of the Ti-6Al-4V alloy’s passive film in a NaCl solution affected the tensile performance during slow strain rate testing (SSRT). Additionally, a high hydrostatic pressure could enhance the SCC of Ti alloys by facilitating the dissolution of Ti and reducing the resistance of the oxide film [4]. The microstructural heterogeneity of welded Ti alloy joints has a remarkable influence on the electrochemical corrosion and mechanical properties of the weldment [5].

Currently, the basic theories for interpreting the SCC mechanism of different material/environmental systems can be classified into two groups, that is, anodic dissolution (AD) and hydrogen embrittlement (HE). On this basis, a variety of specific explanations have been proposed, such as slip dissolution [6], environmentally assisted cracking [7], film-induced cleavage [8] models, hydrogen pressure-induced cracking [9], hydrogen-enhanced localized plasticity (HELP) [10], hydrogen-enhanced decohesion (HEDE) [11], and mixed AD and HE [12] mechanisms. The aging effect of different potentials have differed considerably, corresponding to different SCC mechanisms. Consequently, tensile tests at different potentials have been helpful in exciting different SCC mechanisms to help us examine the SCC behavior of materials under specific service conditions. For the assessment of SCC susceptibility, the ISO 7539-7 and ASTM G129 standards provide testing method guidelines. In recent years, Liu et al. [13,14] proposed a nonsteady electrochemical concept to understand and evaluate the SCC mechanism and susceptibility, aimed at establishing a quantitative model for the rapid evaluation and field detection of SCC susceptibility.

In this study, the SCC susceptibility of a gas-tungsten-arc-welded (GTAW) Ti-6Al-3Nb-2Zr-1Mo (Ti6321) Ti alloy manufactured for deep-sea engineering was investigated. Using electrochemical measurements, a quantitative evaluation model of the SCC susceptibility in deep-sea environments was established. Based on the model, the SCC mechanism in different potential regions is discussed, the results being compared with those from a simulated shallow-sea environment.

## 2. Experimental Methods

### 2.1. Materials and Solution

The as-received Ti6321 alloy was fabricated by Luoyang Ship Material Research Institute (LSMRI, Luoyang, China). The base material (BM) used was a 20 mm thick plate with the main chemical composition (wt. %) of Al 5.5~6.5, Nb 2.5~3.5, Zr 1.5~2.5, Mo 0.6~1.5, Ti bal. It was prepared by arc melting, hot forging, hot rolling, and heat treatment at 980 °C for 1.5 h followed by air cooling to obtain a duplex microstructure. The yield stress of the BM was approximately 800 MPa. To prepare butt-welded joints, GTAW was performed on the plate under argon protection, with the weld direction perpendicular to the rolling direction. The main chemical composition (wt. %) of the filling material was Al 4.0~5.5, Nb 2.0~3.5, Zr 1.0~2.5, Mo 0.5~2.0, Ti bal. After welding, the plate was annealed at 650 °C for 1 h to relieve stress. The welded plate is shown in Figure 1.

The microstructures of the BM, heat-affected zone (HAZ), and weld material (WM) of the weldment were analyzed, as shown in Figure 2. The Ti6321 alloy is a near-α Ti alloy with a duplex microstructure. The ellipsoidal primary α phase is uniformly dispersed in the transformed β matrix (Figure 2a), which provides good corrosion resistance and mechanical performance.

Affected by the welding heat input, the HAZ and WM were heated to the α→β transition temperature (ca. 1020 °C). After air cooling, the α phase precipitates from the prior-β grains, and coarse martensite α’ grains are formed. In α’ grains, secondary α and residual β are stacked and intertwined with each other. The prior-β grain boundary can be clearly seen in the HAZ. During rapid cooling, the growth of secondary α in the HAZ is inhibited, leading to a fine substructure of acicular α and β in the α’ lath [15] (Figure 2b). Owing to the different orientations of the acicular α colonies, the HAZ exhibits a slightly higher SCC resistance and more circuitous crack propagation [16]. However, in the WM, a plate-like secondary α is derived during relatively slow cooling (Figure 2c), which can lead to changes in the mechanical performance.

The specimens used for SSRT were cut from a GTAW Ti6321 plate, with the welded joint located at the center, as shown in Figure 3. A copper wire was connected to one clamping end of the specimen, both ends being coated with silicone rubber, exposing the parallel segment. The rectangular pieces used for electrochemical measurements were machined along the rolling direction of the plate surface and had the same BM and welded joint area ratio as the SSRT specimens. The backs of the specimens were welded using a copper wire before being embedded in epoxy resin, leaving a surface area of 1.5 cm^2^ for the tests. Before testing, all the specimens were sequentially ground with silicon carbide to 1500 grit, degreased in ethanol, ultrasonically washed in distilled water, and dried in cold air.

Natural seawater was used as the test solution, the solution conditions of which are listed in Table 1; they were defined based on the annual mean value of the key environmental factors of the South China Sea, principally its 1500 m deep sea and shallow sea. The data for the environmental factors were obtained from our natural deep-sea experiments. The solution conditions were automatically controlled using a Cortest environmental testing device (Cortest Inc., Willoughby, OH, USA), the low temperature being controlled using a cooling water circulator in the test chamber. A low dissolved oxygen concentration was maintained by purging with N_2_ in the solution reservoir. High pressure was simulated by pumping the solution from the reservoir to the test chamber.

### 2.2. Slow Strain Rate Testing

In situ SSRT was conducted based on the ISO 7539-7 standard at different applied potentials in the same testing device at a strain rate of 10^−6^ s^−1^. After testing, the fracture area and fracture strength were analyzed. The SCC susceptibility reflected in the plasticity loss and strength loss can be calculated using Equation (1), as follows:(1)$$\begin{array}{l} I_{\psi }=\frac{\psi_{a}-\psi_{e}}{\psi_{a}}\times 100\%\\ I_{\sigma }=\frac{\sigma_{a}-\sigma_{e}}{\sigma_{a}}\times 100\% \end{array}$$
where *ψ* is the reduction ratio of the fracture area and can be expressed by Equation (2). *S*_0_ and *S*_1_ refer to the cross-sectional areas before and after the SSRT, respectively.
(2)$$\psi =\frac{S_{0}-S_{1}}{S_{0}}\times 100\%$$
where *σ* is the tensile strength, which can be obtained from the stress–strain curve. The subscripts a and e represent the results in air and the solution, respectively.

The fracture morphology was observed using SEM. The number and size of secondary cracks on the lateral surface of the fractures were statistically analyzed.

### 2.3. Electrochemical Testing

Electrochemical measurements were performed in simulated environments. A traditional three-electrode system was used with a Solartron ModuLab XM electrochemical workstation (XM-studio ver. 3.4, Ametek Inc., Berwyn, PA, USA), including a platinum plate as the counter electrode, a Ag/AgCl solid-state electrode as the reference electrode, and an electrochemical specimen as the working electrode. Before testing, the working electrode was polarized at −1 V for 5 min to ensure a repeatable surface state. Potentiodynamic polarization was conducted within the potential range from −2 V to 2 V at scanning rates of 20 and 2000 mV/min, respectively. Dynamic electrochemical impedance spectroscopy (DEIS, XM-studio ver. 3.4, Ametek Inc., Berwyn, PA, USA) was performed from −1.5 V to 2 V using a potential step of 100 mV, allowing tracing of the corrosion process dynamics based on the evaluation of electrical parameters of the equivalent circuit. The frequency was controlled from 100 kHz to 10 mHz by imposing a sinusoidal voltage amplitude of 10 mV.

## 3. Results and Discussion

### 3.1. Selection of Applied Potential for SSRT

The potentiodynamic polarization curves at low scanning rates in different simulated environments are shown in Figure 4. An obvious passive potential region above the transient activation–passivation transition stage can be seen at both depths. The passive current density in the simulated deep-sea environment is one order of magnitude higher than that in the simulated shallow-sea environment, indicating that the passivation behavior and stability of the passive film in the deep sea are substantially worse. The anodic current peak in the passive region indicates the transformation of titanium oxide (TiO_2_) from TiO or Ti_2_O_3_ to TiO_2_ [17]. However, the corrosion potential is more positive in a simulated deep-sea environment. This is because the corrosion potential shifts positively with a decrease in temperature or increase in dissolved oxygen, while the most important factor affecting the corrosion potential is temperature (accounting for 89.76%), followed by dissolved oxygen (10.24%) [18].

The cathodic Tafel slope (*b*_c_) increases from 118 mV/dec (shallow sea) to 192 mV/dec (deep sea), indicating that the oxygen reduction was suppressed owing to the low dissolved oxygen concentration of the deep sea. However, the limit diffusion region is narrow, and the hydrogen evolution potential is positive in the simulated deep-sea environment. This suggests an increased tendency for hydrogen evolution in the deep sea under high hydrostatic pressure [19]. It can be deduced that the increase in hydrogen evolution overcompensates for the decrease in oxygen reduction. Consequently, the cathodic current density in the simulated deep-sea environment is higher than that in the shallow sea.

Based on the classical theory of Parkins [20,21], potentiodynamic polarization at different potential scanning rates can be used to evaluate the SCC susceptibility determined by AD. Scanning at a slow rate ensures a quasi-steady state and sufficient polarization at the metal surface, which manifests the electrochemical characteristics of the crack walls. Under rapid polarization, the influence of the passive film growth can be eliminated as much as possible, guaranteeing intensive AD, reflecting the electrochemical characteristics of the crack tips. Liu et al. [22,23] extended the scope of this theory to a strong cathodic polarization condition. In this study, potentiodynamic polarization was performed at different scanning rates. Based on the three characteristic potentials found in the curves—that is, the passive potential and null-current potential at a slow scanning rate and the null-current potential at a fast scanning rate—four regions corresponding to different SCC mechanisms could be generally divided, as shown in Figure 4. The Ti6321 welded joints are in a passive state in Region I, exhibiting a narrow active dissolution region in Region II. In Region IV, hydrogen evolution occurs intensively. Region III is considered a transition region of the dominant mechanism from AD to HE. Therefore, several aging potentials for SSRT were selected within the divided potential regions. In the simulated deep-sea environment, the potentials of −1.3, −1, −0.9, −0.8, −0.46, *E*_corr_ (−0.24), 0.1, 0.8, and 1.5 V were chosen. In the simulated shallow-sea environment, the experiments were designed at −1.3, −1, −0.8, −0.6, *E*_corr_ (−0.35), −0.15, 0.8, 1.5, and 2 V, respectively. A tensile test was also performed in air for comparison.

### 3.2. Evaluation of SCC Susceptibility Using Standard SSRT

#### 3.2.1. Stress–Strain Curve Analysis

The stress–strain curves at various potentials in the simulated environments and the SCC susceptibilities reflected by *I_ψ_* and *I_σ_* are shown in Figure 5 and Figure 6, respectively. There is an increase in SCC susceptibility in a simulated deep–sea environment compared to that in a simulated shallow–sea environment. The applied potentials have a major influence on the SCC behavior of the Ti6321 welded joints. The alloy shows a relatively low sensitivity to SCC in Regions II and III. However, at the strong polarization potentials of Regions I and IV, the plasticity decreases sharply. Notably, the strength of the alloy is less influenced by the applied potentials. Consequently, it is necessary to inspect the fracture morphology to confirm the potential–dependent SCC susceptibility.

#### 3.2.2. Fracture Morphology Observations

Figure 7 and Figure 8 show the fracture morphologies in a simulated deep-sea environment at various potentials. The Ti6321 welded joints exhibit high brittle cracking characteristics at strong polarization potentials, manifested in an inclined fracture morphology with wide and deep secondary cracks (Figure 8(a2,c2)). In the potential range of −1.3 to −0.9 V, the fracture surface is characterized by quasi-cleavage and microcracks, the brittleness increasing with a negative shift in potential (Figure 7(a1–c1,a2–c2)). When the potential shifts from 0.8 to 1.5 V, slip steps can be seen in the fracture surface and the dimples degenerate into a river-like pattern (Figure 7(h1–i1,h2–i2)). However, at weak and moderate polarization potentials of −0.8 to −0.46 V, the fracture exhibits an obvious necking phenomenon with few microcracks (Figure 8(b2)), showing a typical dimple morphology (Figure 7(d1–g1,d2–g2)). However, Ti6321 welded joints have slightly higher SCC susceptibility at −0.46 and 0.1 V, manifested in the shallow dimples or ambiguous microcracks, consistent with the potential-dependent SCC susceptibilities reflected by *I_ψ_*. It can also be seen from the fracture morphologies that cracks appear along the slip steps or slip bands at *E*_corr_ and anodic polarization potentials. However, only a slight plastic deformation can be seen before cracking in the case of cathodic polarization. This suggests that plastic deformation is the forerunner for crack initiation, whether AD or HE is the dominant mechanism. However, hydrogen-induced delayed fractures hinder plastic deformation, which substantially reduces the critical fracture strength [24]. Consequently, *I_σ_* is less affected at anodic potentials but decreases to a certain extent when the cathodic polarization is strong. Moreover, by examining the metallography at the main crack sites, all specimens can be seen to be fractured in the WM (Figure 8). This is because the thick secondary α phase in the coarse martensite α’ grains (Figure 2c) softens the microstructure and reduces its strength, leading to localization of the stress field and plastic deformation in the α phase. Therefore, prior to the BM and WM, crack initiation occurs in the WM and propagates through α.

Similar phenomena can be seen in simulated shallow-sea environments at various potentials (Figure 9 and Figure 10). All Ti6321 welded joint specimens eventually fracture in the WM with higher SCC susceptibility at strong polarization levels and lower SCC susceptibility at weak polarization potentials. However, the SCC susceptibility reflected by the fracture morphology is lower than that of the simulated deep-sea environment. In particular, the development of the secondary crack on the lateral surface of the fracture is conservative regardless of the crack length or crack size (Figure 10). Moreover, other ductile characteristics can be seen in the fracture surface, especially at −0.8 V to *E*_corr_ (Figure 9(d1–g1,d2–g2)). The fracture also exhibits considerable brittleness with obvious microcracks at strong polarization potentials (Figure 9(a1–c1,h1–i1,a2–c2,h2–i2).

To further examine the differences in SCC susceptibility in different simulated environments, the cracks number and size were statistically analyzed.

#### 3.2.3. Statistical Analysis of the Number and Size of Cracks

Figure 11 shows the occurrence ratio of secondary cracks at various potentials in different areas of the Ti6321 welded joints in simulated deep-sea and shallow-sea environments. The number of secondary cracks accounts for approximately 60, 25, and 15% of the cracks in the WM, HAZ, and BM, respectively. Because the secondary crack is accompanied by main crack propagation, the result proves that the WM is the final fracture region. At some special potential, such as −0.8 V and *E*_corr_, there are more opportunities for crack initiation during sufficient plastic deformation (owing to the low SCC susceptibility). Consequently, a relatively high ratio can be observed.

The secondary crack length is another useful parameter for assessing SCC susceptibility. Therefore, the distribution of the secondary crack lengths was also statistically analyzed and fitted using a Gaussian function [25]. The results are presented in Figure 12 and Figure 13, and Table 2. The variation law of the secondary crack’s most probable length with applied potential is the same as that of *I_ψ_*. The largest cracks emerge in region IV, followed by those in Region I. Relatively small cracks appear in Regions II and III, especially at −0.8 V and *E*_corr_. The deep-sea environment tends to produce larger cracks than the shallow-sea environment, leading to higher SCC susceptibility. Meanwhile, the distribution of secondary cracks in Region IV is more concentrated than in other regions, indicating more stress and strain concentrations due to higher embrittlement.

### 3.3. Evaluation of SCC Susceptibility Using an Electrochemical Method

#### 3.3.1. Potentiodynamic Polarization Analysis

According to the literature [14], the SCC susceptibility and mechanism in Regions II–IV can be illustrated by means of the current density of the polarization curves at different scanning rates, active dissolution playing a key role in Region II. The current density at a fast-scanning rate (*i*_f_) is always larger than that at a low scanning rate (*i*_s_). This shows that the metal dissolution rate of the crack tip is always higher than that of the non-crack tip—that is, the crack tip grows in the form of AD, the SCC being controlled by the AD mechanism. In Region III, *i*_s_ converted into a cathodic current, which indicates that the crack tip is still affected by AD, while the non-crack tip area is safeguarded by cathodic protection, the gap between *i*_f_ and *i*_s_ decreasing with a negative shift of the potential, indicating the weakening effect of AD. In Region IV, *i*_f_ is also changed to a cathodic current, and the cathodic polarization curves almost coincide, indicating that the effect of AD can be ignored, the hydrogen evolution becoming the main cathodic reaction, the SCC being controlled by HE. With a negative shift of the potential, the hydrogen evolution current increases, the effect of HE increasing too. In Region I, the SCC is strongly affected by the electrochemical competition between passive film growth and dissolution. Consequently, the SCC behavior can no longer be explained by AD alone.

#### 3.3.2. Electrochemical Impedance Spectroscopy Analysis

To further characterize the SCC susceptibility of the Ti6321 welded joints in Region I, DEIS measurements were performed. The plots in the simulated deep- and shallow-sea environments are shown in Figure 14 and Figure 15, respectively. The equivalent circuit shown in Figure 16 was employed to fit the spectra, in which *R*_s_ is the solution resistance, *Q*_f_ and *R*_f_ denote the capacitance and resistance of the corrosion product film or passive film, respectively, *Q*_dl_ and *R*_t_ are the capacitance of the electric double layer and charge transfer resistance, respectively, *Q* representing the constant phase element.

In the passivation region, the SCC behavior is related to the electrochemical reaction process at the solution/passive film interface and the passive film/substrate interface. It has been shown that oxygen vacancies are the main point defects in the passive film of the Ti6321 alloy, which may constitute a channel for electrolyte transfer and control the electrochemical equilibrium of passive film dissolution and growth [17]. Consequently, *R*_f_ and *R*_t_ reflect the sensitivity to SCC in the intact passive film and at the defect point, respectively. Therefore, when *R_s_* can be ignored, the reciprocal of the polarization resistance (1/*R*_p_), where *R*_p_
*= R*_f_
*+ R*_t_*,* can be used to express SCC susceptibility. The relationship between lg(1/*R*_p_) and the applied potential (*E*) is shown in Figure 14 and Figure 15**,** respectively. It can be seen that the variation law of 1/*R*_p_ with *E* is consistent with the SCC susceptibility obtained by SSRT, especially in Region I. Therefore, it is reasonable to use 1/*R*_p_ to reflect the SCC susceptibility affected by the imposed passivation. Generally, the value of 1/*R*_p_ in a simulated deep-sea environment is one order of magnitude higher than that in a simulated shallow-sea environment, indicating that the corrosion resistance of the passive film of the alloy deteriorates in the deep sea. This is consistent with the polarization measurement results.

#### 3.3.3. Electrochemical Evaluation Model

Based on the above discussion, when *I_ψ_* is used as a reference, by applying the literature model [22,23] and supplementing it in the passivation region, the criterion of Equation (3) can be used to evaluate the potential-dependent SCC susceptibility, where *k*_f_, *k*_AD_, and *k*_HE_ are constants related to the material, solution, and electrochemical parameters, respectively, *I*_a_ is a constant related to the electrochemical impedance, *I*_b_ is the nominal SCC susceptibility when *i*_f_ = *i*_s_, *I*_c_ is the nominal susceptibility when *i*_s_ = *i*_corr_, *I*_d_ is a constant related to AD and HE. |*i*_s_| characterizes the effect of the hydrogen evolution, the HE effect being enhanced with an increase in |*i*_s_|. *i*_f_·|(*i*_f_ − *i*_s_)/*i*_s_| represents the crack growth rate, the higher the value, the faster the crack growth. lg(1/*R*_p_) represents the SCC susceptibility related to the state of the passive film. The tendency of film-assisted cracking increases with an increase in lg(1/*R*_p_).
(3)$$I_{\psi }=\left\{\begin{array}{ll} k_{f}\cdot \mathrm{lg}\frac{1}{R_{p}}+I_{a} & \mathrm{Region}\;I\\ k_{\mathrm{AD}}\cdot i_{f}\cdot \left|\frac{i_{f}-i_{s}}{i_{s}}\right|+I_{b} & \mathrm{Region}\;\mathrm{II}\\ k_{\mathrm{HE}}\cdot \left|i_{s}\right|+k_{\mathrm{AD}}\cdot i_{f}\cdot \left|\frac{i_{f}-i_{s}}{i_{s}}\right|+I_{c} & \mathrm{Region}\;\mathrm{III}\\ k_{\mathrm{HE}}\cdot \left|i_{s}\right|+I_{d} & \mathrm{Region}\;\mathrm{IV} \end{array}\right.$$

The coefficients in Equation (3) were obtained by applying the *I_ψ_* obtained in the SSRT test and the *i*_f_, *i*_s_ or *R*_p_ at a corresponding potential. The calculation method and results are shown in Table 3.

The evaluation models for the SCC susceptibility of Ti6321 welded joints in simulated deep-sea and shallow-sea environments can be expressed as shown in Equation (4), the characterization curves being shown in Figure 17. The calculated data are in good agreement with the measured data and can predict the SCC susceptibility under different mechanisms.

In Regions I and II, the SCC behaviors of the Ti6321 welded joints depend on the electrochemical properties of the passive film and are driven by dislocation slip under stress. The low dissolved oxygen and low temperature of the deep sea retards the formation of chemically stable components in passive films [26], for example, TiO_2_, Nb_2_O_5_, and ZrO_2_, which could reduce Cl^−^ ingress and increase the structural integrity of the passive film [27]. Therefore, the passive current density increases (Figure 4). Accompanied by dislocation slip motion (see that the fracture features are indicative of pronounced local plastic deformation, as shown in Figure 7 and Figure 8), the passive film ruptures at the weak points. Subsequently, the dissolution and growth equilibrium of the passive film at the film/substrate interface is destroyed, leading to intensive AD of the exposed fresh metal under the promotion of hydrostatic pressure on the adsorption and permeation of Cl^−^ [28], as well as the depassivation effect of H^+^ [29], until a new balance is rebuilt. Repeated slip dissolution leads to cracking and final fracturing. A schematic of this mechanism [30] is shown in Figure 18a.


(4)
$$\begin{array}{cc} I_{\psi }=\left\{\begin{array}{cc} 23.77\cdot \mathrm{lg}\frac{1}{R_{p}}+99.66 & \;\mathrm{Region}\;I\\ -17.85\cdot i_{f}\cdot \left|\frac{i_{f}-i_{s}}{i_{s}}\right|+17.27 & \;\mathrm{Region}\;\mathrm{II}\\ -4.531\times 10^{4}\cdot \left|i_{s}\right|-38.80\cdot i_{f}\cdot \left|\frac{i_{f}-i_{s}}{i_{s}}\right|+25.29 & \;\mathrm{Region}\;\mathrm{III}\\ 3.706\times 10^{3}\cdot \left|i_{s}\right|+26.95 & \;\mathrm{Region}\;\mathrm{IV} \end{array}\right. & \mathrm{for}\;\mathrm{deep}\;\mathrm{sea}\\ I_{\psi }=\left\{\begin{array}{cc} 7.95\cdot \mathrm{lg}\frac{1}{R_{p}}+57.65 & \mathrm{Region}\;I\\ 5.878\times 10^{3}\cdot i_{f}\cdot \left|\frac{i_{f}-i_{s}}{i_{s}}\right|+28.11 & \mathrm{Region}\;\mathrm{II}\\ -1.314\times 10^{6}\cdot \left|i_{s}\right|-6.669\times 10^{3}\cdot i_{f}\cdot \left|\frac{i_{f}-i_{s}}{i_{s}}\right|+31.32 & \mathrm{Region}\;\mathrm{III}\\ 6.008\times 10^{3}\cdot \left|i_{s}\right|+28.06 & \mathrm{Region}\;\mathrm{IV} \end{array}\right. & \mathrm{for}\;\mathrm{shallow}\;\mathrm{sea} \end{array}$$


The vulnerable film in the deep sea makes the material more sensitive to SCC. With a positive shift of the potential in Region II, Ti6321 welded joints are temporarily activated, the effect of AD being enhanced. As the potential continues to shift positively to Region I, electrochemical passivation increases the stability and intactness of the passive film, reducing the tendency of crack initiation. However, once the crack is initiated under stress, the rapid thickening of the passive film on the crack walls can produce a wedge force promoting the cleavage of the crack-tip metal [31]. Consequently, the SCC susceptibility at passive potentials is higher than that in the naturally passivated state. With a positive shift in the potential, the dissolution rate of the passive film exceeds its growth rate, the film defects increase, and *R*_p_ decreases (Figure 14 and Figure 15), leading to an increase in SCC susceptibility. When the potential reaches approximately 1.5 V, TiO or Ti_2_O_3_ starts to oxidize to TiO_2_ [26], the SCC susceptibility decreasing again.

Region III refers to the regular potential region employed for cathodic protection of marine engineering materials and alloys [32]. According to mixed-potential theory [33], *R*_p_ is equivalent to the parallel connection of the anodic charge transfer resistance (*R*_pa_) and cathodic charge transfer resistance (*R*_pc_). With a negative shift in the potential, *R*_pa_ increases gradually, implying a gradually reducing AD effect. By contrast, *R*_pc_ decreases rapidly at first with cathodic oxygen reduction, before decreasing slowly when the limit diffusion of oxygen is achieved, and again decreasing rapidly with the occurrence of hydrogen evolution. In general, *R*_p_ decreases first, before increasing, and then decreasing again, as shown in Figure 14 and Figure 15, the SCC susceptibility changing accordingly. At approximately −0.8 V, the effect of AD is negligible, and hydrogen evolution has not yet occurred [12,22]. Consequently, the specimen is effectively safeguarded by cathodic protection, the SCC susceptibility reaching a minimum in Region III.

In Region IV, the cathodic process is controlled by hydrogen evolution. With a negative shift of the potential, *R*_p_ decreases, the current density increasing rapidly, suggesting an enhanced HE effect. Hydrogen can change the composition of the passive film and produce hydrides [34], resulting in brittle cracking under stress. Hydrostatic pressure promotes the penetration of adsorbed hydrogen and the formation of hydrides at the α/β interface [4]. The solid solubility of hydrogen in the α phase is limited, which is more likely to cause lattice distortion of the α phase and make the microstructure more unstable [35]. With reference to the findings of Xie et al. [36], hydrogenated vacancies facilitate dislocation motion and lock the accumulated dislocation at the crack tip to produce a local high-stress field. While most of the hydrogen may drain into the bulk owing to its high solubility in the β phase, there is still a small amount of hydrogen retained in the stress field of the crack tip [37]. The hydrogen at trap sites or the resultant hydride in the crack tip can internally pressurize the material [38]. When the HELP reaches a critical value, the HEDE is pronounced, leading to crack propagation across the α phase. A schematic of this mechanism is shown in Figure 18b. Furthermore, Ti6321 welded joints exhibit substantial HE in a simulated deep-sea environment with a more positive upper-boundary potential (critical susceptibility potential of the HE effect, which is defined by 25% of *I_ψ_*) in Region IV, as well as much larger secondary cracks (Table 2). This can be attributed to the significant increase in the trend of hydrogen evolution in the deep sea, as indicated by the polarization measurements.

## 4. Conclusions

(1)Ti6321 GTAW-welded joints were fractured in the WM because the coarse martensite grains softened the microstructure and reduced its strength, leading to the localization of the stress field and plastic deformation in the thick secondary α phase.(2)Ti6321 welded joint exhibited higher stress corrosion susceptibility in a simulated deep-sea environment and at a strong polarization level owing to the diminishing protection of the passive film under passivation inhibition and the enhancement of the hydrogen effect. The applied potential affected the plasticity of the material but had little influence on its strength.(3)The potential-dependent SCC mechanism could be classified theoretically using potentiodynamic polarization at different scanning rates and dynamic electrochemical impedance spectroscopy measurements, that is, film-induced cleavage in the passivation region, slip dissolution in the activation region, cathodic protection in the transition region, and the hydrogen embrittlement region.(4)Strong polarization significantly increased SCC susceptibility, manifested by a decrease in plasticity. The sensitivity to HE was higher in the deep sea because of the increased tendency for hydrogen evolution. At approximately −0.8 V, the material received the best cathodic protection.(5)This study is of practical value to the deep-sea engineering application of Ti6321 and other titanium alloys, especially in the prediction of SCC susceptibility, the judgement of SCC mechanisms and the design of cathodic protection.

## Figures and Tables

**Figure 1 materials-15-03201-f001:**
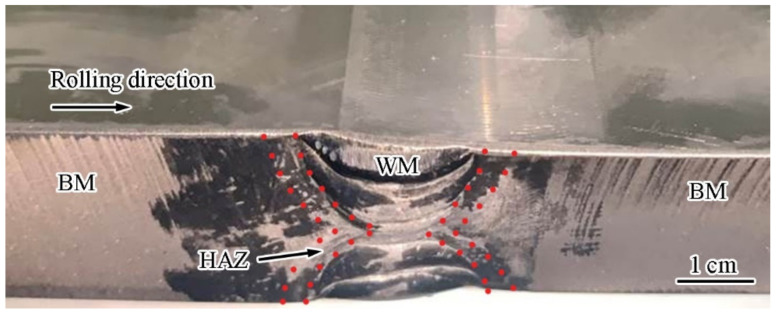
Ti6321 alloy GTAW plate.

**Figure 2 materials-15-03201-f002:**
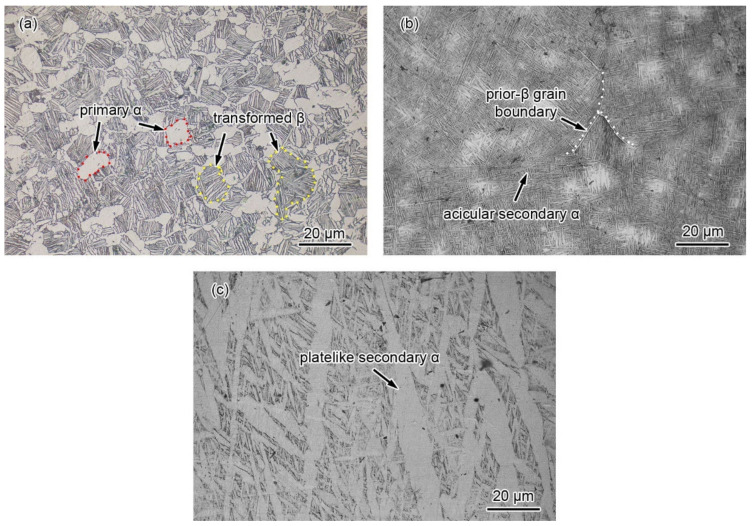
Microstructure of the base material (**a**), heat–affected zone (**b**), and weld material (**c**) of the weldment.

**Figure 3 materials-15-03201-f003:**
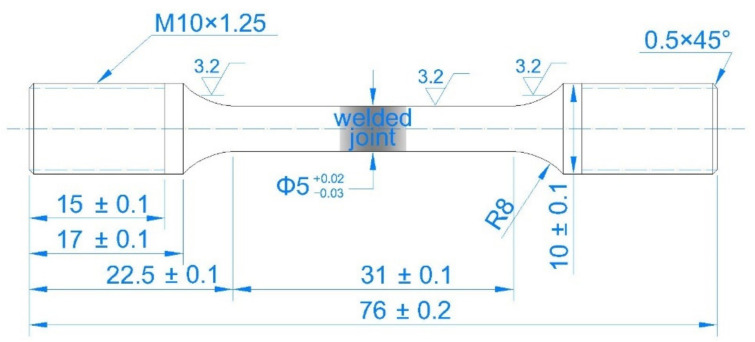
Welding specimen used for slow strain rate testing (units in mm).

**Figure 4 materials-15-03201-f004:**
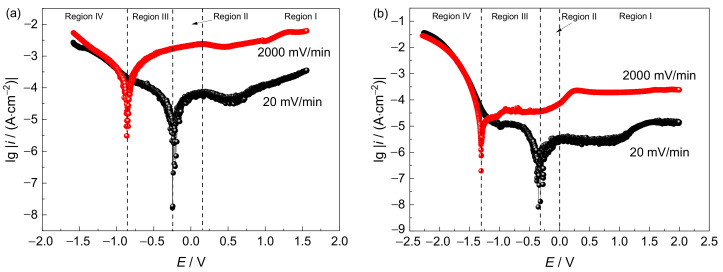
Potentiodynamic polarization curves at different scanning rates in simulated deep−sea (**a**) and shallow−sea (**b**) environments.

**Figure 5 materials-15-03201-f005:**
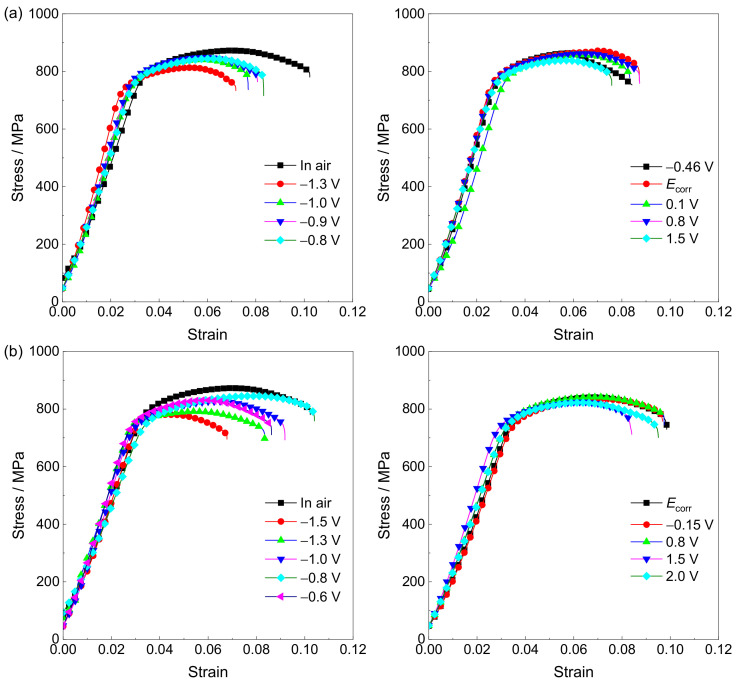
Stress–strain curves at various potentials in simulated deep–sea (**a**) and shallow–sea (**b**) environments.

**Figure 6 materials-15-03201-f006:**
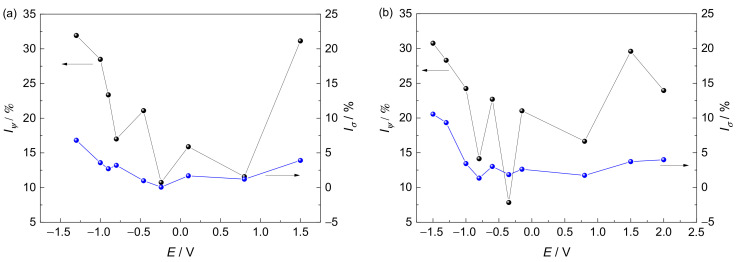
Reduction-in-area loss and fracture–stress loss at various potentials in simulated deep–sea (**a**) and shallow–sea (**b**) environments.

**Figure 7 materials-15-03201-f007:**
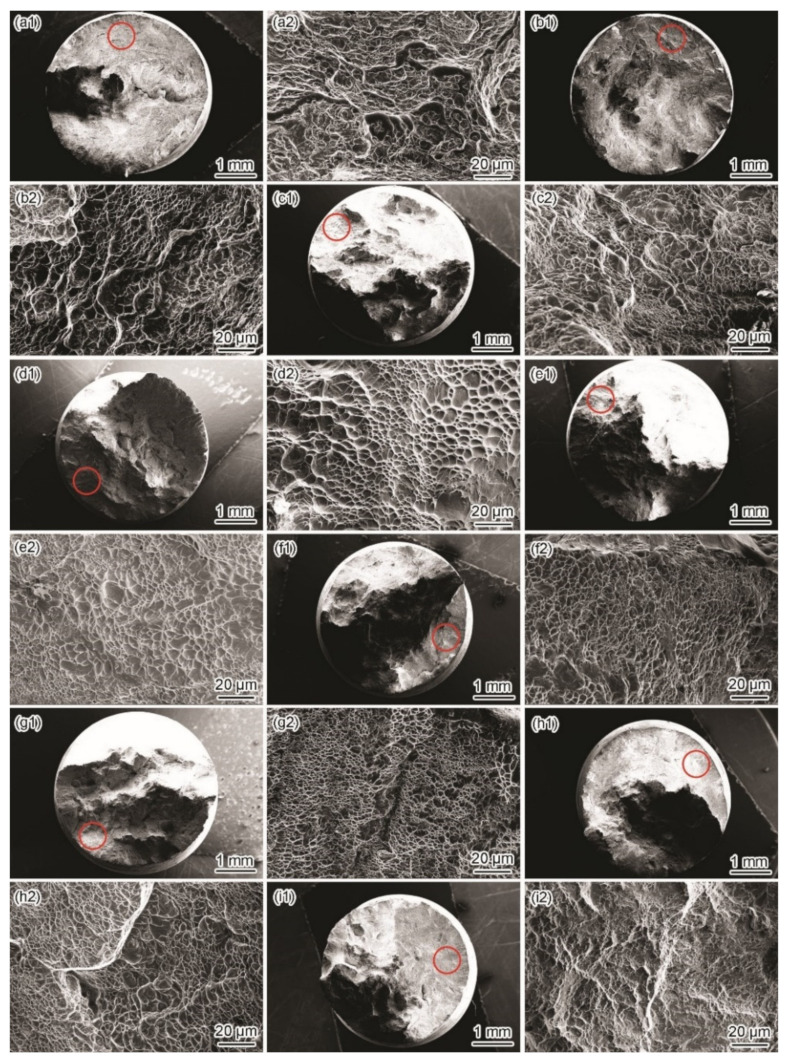
Fracture surface morphologies in a simulated deep–sea environment. Macromorphology of fracture surface at −1.3 V (**a1**), −1 V (**b1**), −0.9 V (**c1**), −0.8 V (**d1**), −0.46 V (**e1**), *E*_corr_ (**f1**), 0.1 V (**g1**), 0.8 V (**h1**), and 1.5 V (**i1**); micromorphology at crack initiation site marked with red circle in (**a1**–**i1**) at −1.3 V (**a2**), −1 V (**b2**), −0.9 V (**c2**), −0.8 V (**d2**), −0.46 V (**e2**), *E*_corr_ (**f2**), 0.1 V (**g2**), 0.8 V (**h2**), and 1.5 V (**i2**).

**Figure 8 materials-15-03201-f008:**
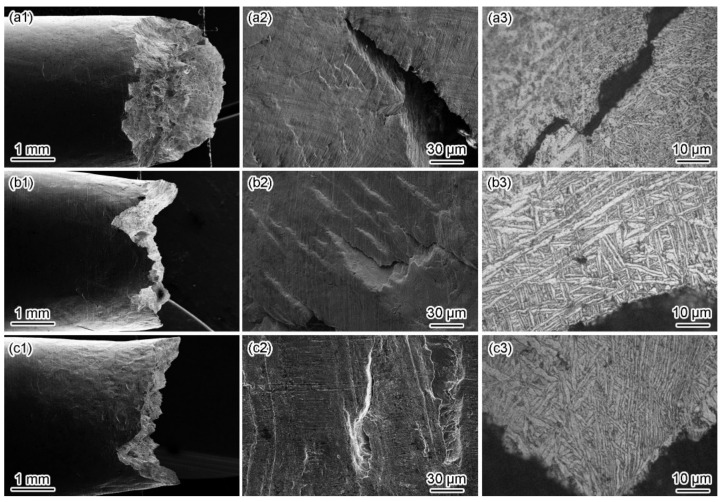
Fracture side morphologies in a simulated deep−sea environment. Macromorphology of fracture side at −1.3 V (**a1**), *E*_corr_ (**b1**), and 1.5 V (**c1**); micromorphology of secondary crack at −1.3 V (**a2**), *E*_corr_ (**b2**), and 1.5 V (**c2**); metallography of main crack at −1.3 V (**a3**), *E*_corr_ (**b3**), and 1.5 V (**c3**).

**Figure 9 materials-15-03201-f009:**
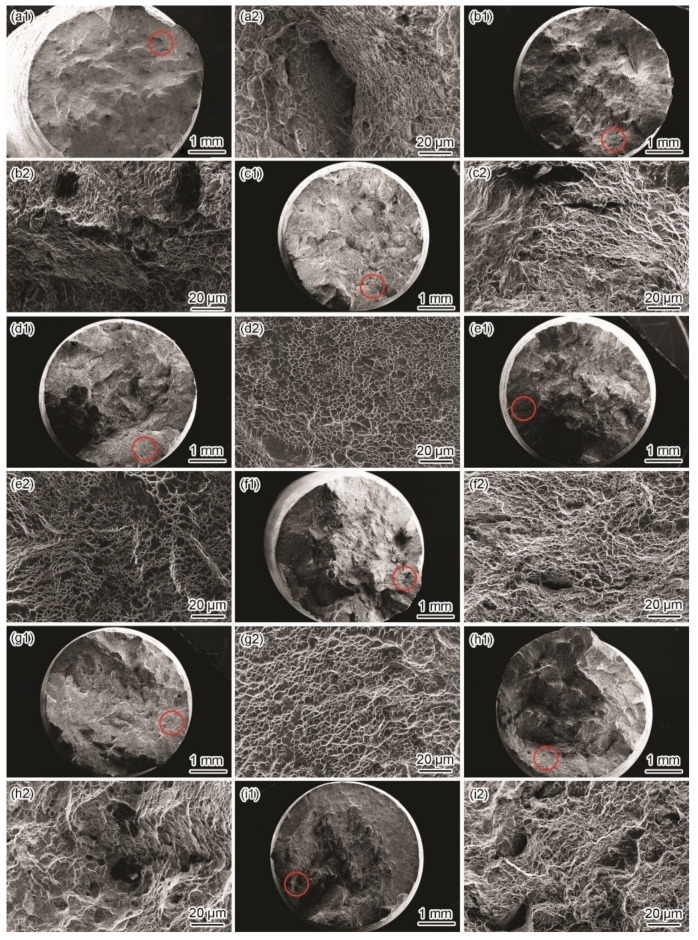
Fracture surface morphologies in a simulated shallow−sea environment. Macromorphology of fracture surface at −1.5 V (**a1**), −1.3 V (**b1**), −1 V (**c1**), −0.8 V (**d1**), −0.6 V (**e1**), −0.15 V (**f1**), *E*_corr_ (**g1**), 0.8 V (**h1**), and 1.5 V (**i1**); micromorphology at crack initiation site marked with red circle in (**a1**–**i1**) at −1.5 V (**a2**), −1.3 V (**b2**), −1 V (**c2**), −0.8 V (**d2**), −0.6 V (**e2**), −0.15 V (**f2**), *E*_corr_ (**g2**), 0.8 V (**h2**), and 1.5 V (**i2**).

**Figure 10 materials-15-03201-f010:**
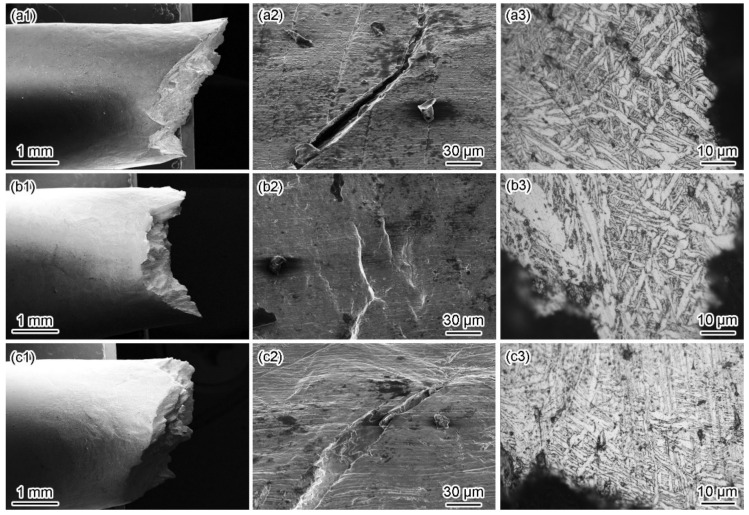
Fracture side morphologies in simulated a shallow−sea environment at −1.5 V (**a**), *E*_corr_ (**b**), and 1.5 V (**c**). Macromorphology of fracture side at −1.5 V (**a1**), *E*_corr_ (**b1**), and 1.5 V (**c1**); micromorphology of secondary crack at −1.5 V (**a2**), *E*_corr_ (**b2**), and 1.5 V (**c2**); metallography of main crack at −1.5 V (**a3**), *E*_corr_ (**b3**), and 1.5 V (**c3**).

**Figure 11 materials-15-03201-f011:**
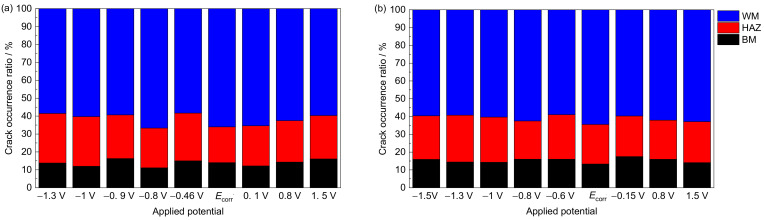
Distribution of secondary crack ratio at various potentials in different areas of welded joints in simulated deep–sea (**a**) and shallow–sea (**b**) environments.

**Figure 12 materials-15-03201-f012:**
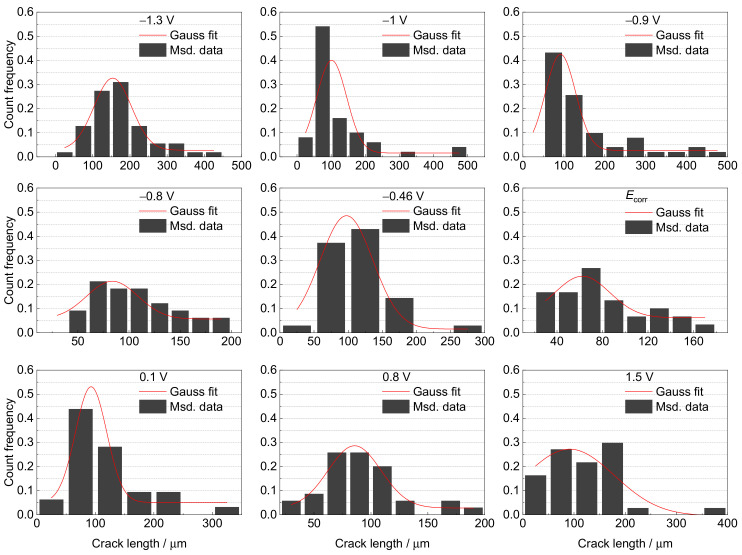
The distribution of secondary crack length near the main crack at various potentials in a simulated deep–sea environment.

**Figure 13 materials-15-03201-f013:**
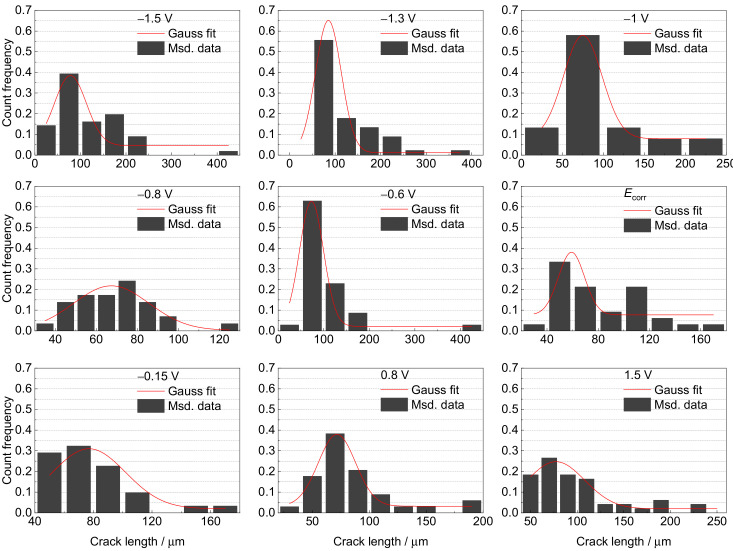
The distribution of secondary crack length near the main crack at various potentials in a simulated shallow–sea environment.

**Figure 14 materials-15-03201-f014:**
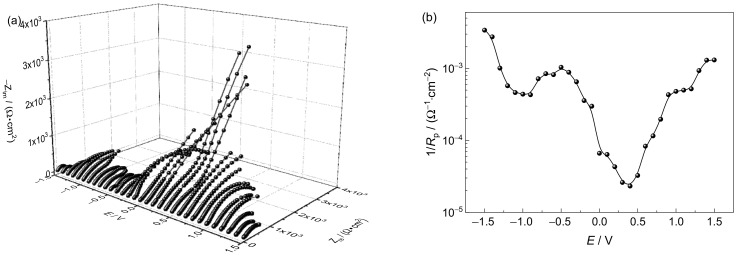
Dynamic electrochemical impedance spectroscopy plot (**a**) and lg(1/*R*_p_)–*E* curve (**b**) in a simulated deep–sea environment.

**Figure 15 materials-15-03201-f015:**
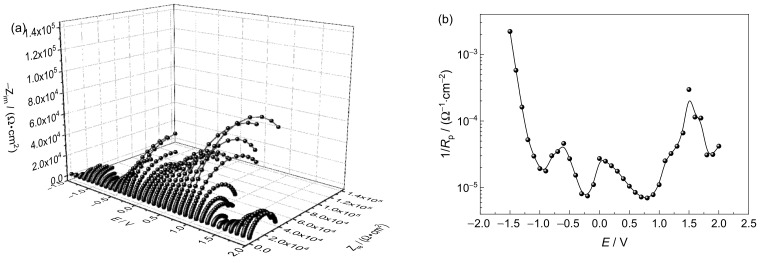
Dynamic electrochemical impedance spectroscopy plot (**a**) and lg(1/*R_p_*)–*E* curve (**b**) in a simulated shallow−sea environment.

**Figure 16 materials-15-03201-f016:**
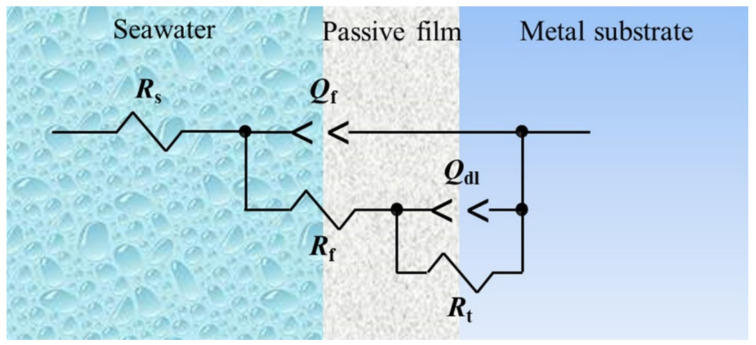
Equivalent circuit used to fit the electrochemical impedance spectroscopy.

**Figure 17 materials-15-03201-f017:**
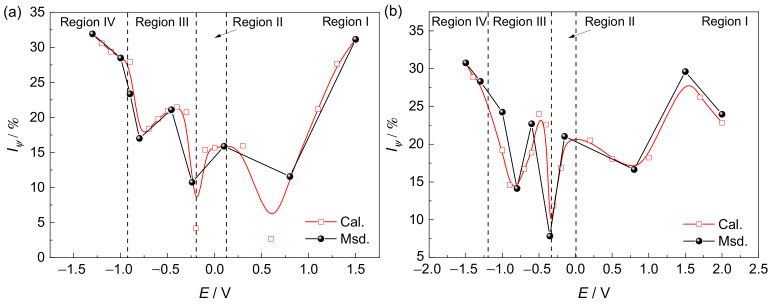
Characterization of stress corrosion cracking susceptibility using the evaluation models in simulated deep−sea (**a**) and shallow−sea (**b**) environments.

**Figure 18 materials-15-03201-f018:**
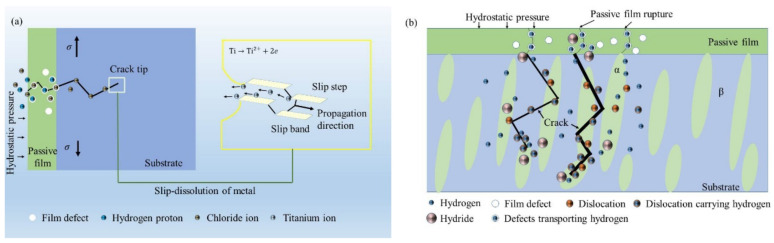
Schematic diagram for slip-dissolution mechanism (**a**) and hydrogen embrittlement mechanism (**b**) in the deep sea.

**Table 1 materials-15-03201-t001:** Test conditions for simulating deep-sea and shallow-sea environments.

Solution Condition	Hydrostatic Pressure (MPa)	Temperature (°C)	Dissolved Oxygen (mg/L)	pH	Salinity (%)
1500 m deep sea	15	28.2	7.4	8.0	3.4
Shallow sea	0.1	2.9	3.5	8.1	3.4

**Table 2 materials-15-03201-t002:** Most probable length of secondary crack at various potentials in simulated deep–sea and shallow–sea environments.

Solution Condition	Most Probable Length of Secondary Crack									
1500 m deep sea	Potential (V)	−1.3	−1	−0.9	−0.8	−0.46	*E* _corr_	0.1	0.8	1.5
	Length (μm)	153.8	101.5	92	83.7	97.5	62.3	92.7	85.6	91.5
Shallow sea	Potential (V)	−1.5	−1.3	−1	−0.8	−0.6	*E* _corr_	−0.15	0.8	1.5
	Length (μm)	78.1	85.3	75.0	67.3	73.2	59.2	77.0	71.6	77.2

**Table 3 materials-15-03201-t003:** The calculation method and results of coefficients of each potential region in simulated deep-sea and shallow-sea environments.

Solution Condition	Potential Region	Corresponding Potential	Input Parameters	Output Coefficients
1500 m deep sea	Region I	0.8, 1.5 V	*I_ψ_*, *R*_p_	*k*_f_ = 23.77, *I*_a_ = 99.66
	Region II	*E*_corr_, 0.1 V	*I_ψ_*_,_ *i*_f_, *i*_s_	*k*_AD_ = −17.85, *I*_b_ = 17.27
	Region III	−0.9, −0.8, −0.46 V	*I_ψ_*_,_ *i*_f_, *i*_s_	*k*_HE_ = −4.531 × 10^4^, *k*_AD_ = −38.80, *I*_c_ = 25.29
	Region IV	−1.3, −1 V	*I_ψ_*_,_ *i*_s_	*k*_HE_ = 3.706 × 10^3^, *I*_d_ = 26.95
Shallow sea	Region I	0.8, 1.5, 2 V	*I_ψ_*, *R*_p_	*k*_f_ = 7.95, *I*_a_ = 57.65
	Region II	*E*_corr_, −0.15 V	*I_ψ_*, *i*_f_, *i*_s_	*k*_AD_ = 5.878 × 10^3^, *I*_b_ = 28.11
	Region III	−1, −0.8, −0.6 V	*I_ψ_*, *i*_f_, *i*_s_	*k*_HE_ = −1.314 × 10^6^, *k*_AD_ = −6.669 × 10^3^, *I*_c_ = 31.32
	Region IV	−1.5, −1.3 V	*I_ψ_*, *i*_s_	*k*_HE_ = 6.008 × 10^3^, *I*_d_ = 28.06

## Data Availability

Data is contained within the article.

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
