# Peer review of "Electrochemical Evaluation of Stress Corrosion Cracking Susceptibility of Ti-6Al-3Nb-2Zr-1Mo Alloy Welded Joint in Simulated Deep-Sea Environment"

_materials, 2022, doi:10.3390/ma15093201_

Round 1
Reviewer 1 Report
The submitted report exhibits the Stress corrosion cracking (SCC) of a gas tungsten arc welded (GTAW) Ti-6Al-3Nb-2Zr-1Mo (Ti6321) alloy influenced by applied potentials in simulated deep-sea and shallow-sea environments are investigated by combining slow strain rate testing (SSRT) with electrochemical measurements. The electrochemical evaluation model for SCC susceptibility of passivation system is established by modifying the existing criteria. The SCC mechanisms are classified and discussed. The results show that the occurrence of fracture of Ti6321 welded joint in the weld material (WM) is attributed to the softening effect of the thick secondary α within coarse-grained martensite. The applied potential mainly affects the plasticity of the material but has little influence on its strength. The potential-dependent SCC mechanism can be classified theoretically by using potentiodynamic polarization at different scanning rates and dynamic electrochemical impedance spectroscopy measurements, which involves film-induced cleavage, slip-dissolution, cathodic protection, and hydrogen embrittlement mechanism. The material shows higher SCC susceptibility in simulated deep-sea environments and at a strong polarization level due to the deteriorated protection of passive film under passivation inhibition and the enhancement of the hydrogen effect. The optimal cathodic potential of Ti6321 welded joint in the simulated marine environment is close to -0.8V.
This research is well arranged, with a sequence of clear ideas and concise writing that fits the research plan and methodology. The literature review is good, and they were able to successfully discuss their progress from both a perspective and an applied perspective. Their chosen method makes this data analysis excellent research and enables them to answer research questions and test their hypotheses. Thus, I strongly recommend this manuscript for publication in the Materials Journal.
Author Response
Dear Reviewer:
We would like to thank you for your careful reading, helpful comments, and constructive suggestions, which have significantly improved the presentation of our manuscript.And we have carefully considered all comments from the reviewers and revised our manuscript accordingly.
The responses to your comments are as follows:
Comment: The submitted report exhibits the Stress corrosion cracking (SCC) of a gas tungsten arc welded (GTAW) Ti-6Al-3Nb-2Zr-1Mo (Ti6321) alloy influenced by applied potentials in simulated deep-sea and shallow-sea environments are investigated by combining slow strain rate testing (SSRT) with electrochemical measurements. The electrochemical evaluation model for SCC susceptibility of passivation system is established by modifying the existing criteria. The SCC mechanisms are classified and discussed. The results show that the occurrence of fracture of Ti6321 welded joint in the weld material (WM) is attributed to the softening effect of the thick secondary α within coarse-grained martensite. The applied potential mainly affects the plasticity of the material but has little influence on its strength. The potential-dependent SCC mechanism can be classified theoretically by using potentiodynamic polarization at different scanning rates and dynamic electrochemical impedance spectroscopy measurements, which involves film-induced cleavage, slip-dissolution, cathodic protection, and hydrogen embrittlement mechanism. The material shows higher SCC susceptibility in simulated deep-sea environments and at a strong polarization level due to the deteriorated protection of passive film under passivation inhibition and the enhancement of the hydrogen effect. The optimal cathodic potential of Ti6321 welded joint in the simulated marine environment is close to -0.8V.
This research is well arranged, with a sequence of clear ideas and concise writing that fits the research plan and methodology. The literature review is good, and they were able to successfully discuss their progress from both a perspective and an applied perspective. Their chosen method makes this data analysis excellent research and enables them to answer research questions and test their hypotheses. Thus, I strongly recommend this manuscript for publication in the Materials Journal.
Response: Much thanks to the reviewer for reading our paper and giving the above positive comments.

Reviewer 2 Report
1) Kindly please the paper format as per journal requirement
2) Kindly please enhance the language standard
3) Refine the abstract section.
4) The specimen material is purchased (or) fabricated??
5) Why two equations have been numbered as 1?
6) section 2.3: last point: is it mHz (or) MHz?
7) Kindly refine the conclusion section?
Author Response
Dear Reviewer:
We would like to thank you for your careful reading, helpful comments, and constructive suggestions, which have significantly improved the presentation of our manuscript. And we have carefully considered all comments from the reviewers and revised our manuscript accordingly. The manuscript has also been double-checked, and the language standard has been improved. We hope our revised manuscript can be accepted for publication.
The responses to your comments are as follows:
Comment 1: Kindly please the paper format as per journal requirement.
Response: We have made changes to the format required by the Materials journal.
Comment 2: Kindly please enhance the language standard.
Response: We have enhanced the language standard.
Comment 3: Refine the abstract section.
Response: We have rewritten the abstract to make it clear, descriptive, and self-explanatory.
Comment 4: The specimen material is purchased (or) fabricated??
Response: The as-received Ti6321 alloy is fabricated by Luoyang Ship Material Research Institute. A description has been added to the first paragraph of section 2.1.
Comment 5: Why two equations have been numbered as 1?
Response: We've checked and corrected all the equation numbers.
Comment 6: section 2.3: last point: is it mHz (or) MHz?
Response: It’s mHz―that is 0.001Hz.
Comment 7: Kindly refine the conclusion section?
Response: We have resummarized the conclusions and hope to leave the reader with a good take-home message.

Reviewer 3 Report
Article (materials-1676196) Title: “
Electrochemical evaluation of stress corrosion cracking susceptibility of Ti-6Al-3Nb-2Zr-1Mo alloy welded joint in simulated deep-sea environment”.
Comments:
- The abstract needs to be rewritten. Novelty should be mentioned in the abstract. Abbreviations should not be used in the abstract. The abstract should be clear in a way that even without reading the whole paper, it still can give some useful information.
- The authors should provide test conditions of salinity (%) and pH for both deep-sea and shallow-sea environments in Table 1. Please revise
Author Response
Dear Reviewer:
We would like to thank you for your careful reading, helpful comments, and constructive suggestions, which have significantly improved the presentation of our manuscript. And we have carefully considered all comments from the reviewers and revised our manuscript accordingly. The manuscript has also been double-checked, and the language standard has been improved.We hope our revised manuscript can be accepted for publication.
The responses to your comments are as follows:
Comment 1: The abstract needs to be rewritten. Novelty should be mentioned in the abstract. Abbreviations should not be used in the abstract. The abstract should be clear in a way that even without reading the whole paper, it still can give some useful information.
Response: We have rewritten the abstract to make it clear, descriptive, and self-explanatory.
Comment 2: The authors should provide test conditions of salinity (%) and pH for both deep-sea and shallow-sea environments in Table 1. Please revise.
Response: The test conditions of salinity and pH have been supplemented in Table 1. The data of environmental factors were obtained from our natural deep-sea experiments. A note has been also added to the last paragraph of section 2.1.

Reviewer 4 Report
The overall paper is interesting and well prepared, however some minor corrections are welcome:
- in section 2 (materials and methods) please include a picture with plate with welds / welded joints to visualize the way that it was prepared, please refer to areas from figure 1 in this visualisation;
- in figures 6-9, please rewrite / redraw the indicators a-h (and scale). In the form presented they are hardly readable;
- in figure 13a - the equivalent circuit shall be placed as independent figure or 13c etc- to make it more clear / readable;
- in part 3.2.3: there are used constants k_f, k_AD, k_HE - please identify them and list in the table (with sources) etc. When analysing equation (3) and (4), above mentioned coefficients differ significantly.
- presented research and results might find its practical application in industry, please indicate clearly recomendations for practical application of the results in the conclusions.
Author Response
Dear Reviewer:
We would like to thank you for your careful reading, helpful comments, and constructive suggestions, which have significantly improved the presentation of our manuscript. And we have carefully considered all comments from the reviewers and revised our manuscript accordingly. The manuscript has also been double-checked, and the language standard has been improved.We hope our revised manuscript can be accepted for publication.
The responses to your comments are as follows:
Comment 1: In section 2 (materials and methods) please include a picture with plate with welds / welded joints to visualize the way that it was prepared, please refer to areas from figure 1 in this visualisation.
Response: In section 2, a picture of GTAW Ti6321 alloy plate has been included as shown in Figure 1.
Comment 2: In figures 6-9, please rewrite / redraw the indicators a-h (and scale). In the form presented they are hardly readable.
Response: We redraw the indicators and scales in Figure 6-9.
Comment 3: In figure 13a - the equivalent circuit shall be placed as independent figure or 13c etc- to make it more clear / readable.
Response: We provide the equivalent circuit as a independent Figure 16.
Comment 4: In part 3.2.3: there are used constants k_f, k_AD, k_HE - please identify them and list in the table (with sources) etc. When analysing equation (3) and (4), above mentioned coefficients differ significantly.
Response: In section 3.3.3., the coefficients in equation (3) were obtained by using the Iψ obtained in the SSRT test and the if, is or Rp at a corresponding potential. The calculation method and results are provided with a new Table 3.
Comment 5: Presented research and results might find its practical application in industry, please indicate clearly recomendations for practical application of the results in the conclusions.
Response: In the conclusions of the revised draft, clear suggestions have been indicated for the practical application of the research results.

Round 2
Reviewer 3 Report
Good effort has been made to address the comments.